# Systematic Investigation of the Role of Molybdenum and Boron in NiCo-Based Alloys for the Oxygen Evolution Reaction

**DOI:** 10.3390/molecules30091971

**Published:** 2025-04-29

**Authors:** Parastoo Mouchani, Donald W. Kirk, Steven J. Thorpe

**Affiliations:** 1Department of Chemical Engineering and Applied Chemistry, University of Toronto, Toronto, ON M5S 3E5, Canada; parastoo.mouchani@utoronto.ca; 2Department of Materials Science and Engineering, University of Toronto, Toronto, ON M5S 3E4, Canada; steven.thorpe@utoronto.ca

**Keywords:** oxygen evolution reaction (OER), electrocatalyst, stability, high-energy surfactant-assisted ball milling (HE-SABM), mechanical alloying, cryo-milling, boron

## Abstract

Quaternary NiCoMoB electrocatalysts exhibited significantly enhanced OER performance compared to their ternary NiCoMo and NiCoB counterparts. An optimal Mo/B ratio of 1 (NiCoMo_y_B_y_) demonstrated a superior OER activity, attributed to a balance between the electronic and structural contributions from Mo and B, maximizing the electrocatalytic site density and activity. NiCoMo_y_B_y_-SA, a nanoparticle version synthesized via a surfactant-assisted method, showed further improved performance. The OER activity was evaluated by comparing overpotentials at 10 mA/cm^2^, with NiCoMo_x_B_1−x_, NiCoMo_y_B_y_, and NiCoMo_y_B_y_-SA exhibiting 293, 284, and 270 mV, respectively. NiCoMo_y_B_y_-SA also demonstrated the lowest onset potential (1.45 V), reflecting a superior efficiency. Chronoamperometry in 1 M pre-electrolyzed KOH at 30 °C highlighted NiCoMo_y_B_y_-SA’s stability, activating within hours at 10 mA/cm^2^ and stabilizing over 7 days. At 50 mA/cm^2^, the overpotential increased minimally (0.02 mV/h over 2 days), and even at 100 mA/cm^2^ for 10 days, the activity declined only slightly, affirming a high stability. These findings demonstrate NiCoMoB electrocatalysts as cost-effective, efficient OER electrocatalysts, advancing sustainable energy technologies.

## 1. Introduction

The development of efficient and cost-effective electrocatalysts for the oxygen evolution reaction (OER) is crucial for advancing renewable energy technologies, such as water splitting, fuel cells, and metal–air batteries [1]. However, the OER is inherently sluggish, requiring high overpotentials to proceed efficiently, making the design of active and stable electrocatalysts a pressing need [2]. As a result, the development of stable and active electrocatalysts is essential to reduce the energy losses associated with this reaction. Transition-metal-based alloys, particularly those involving nickel and cobalt, have emerged as promising candidates for the overall water splitting reaction due to their favorable electronic properties, electrocatalytic activities, and relatively low cost compared to noble metal electrocatalysts such as platinum and iridium [3,4,5].

Recent research efforts have increasingly focused on enhancing the performance of Ni-Co alloys by incorporating additional metal elements such as Fe [6], Cr [7], Mn [8], W [9], Mo [10,11,12], nonmetals (C, N, P, S, and Se), and metalloids (As, Te, and B [13,14,15], which have shown potential in improving both electrocatalytic activity and stability.

Among these elements, the incorporation of Mo into Ni-Co alloys significantly enhances the electrocatalytic performance of these materials for the OER [16]. One of the primary benefits is the alteration of the electronic structure, which modulates the electronic properties of the electrocatalyst [17]. This modification optimizes the adsorption and desorption energies of active species during the OER, thereby facilitating more efficient electron transfer and reducing the overpotential required for the reaction [6,18]. Mo also contributes to the formation of Mo-rich phases and mixed oxides such as NiMoO_4_ and CoMoO_4_, which provide abundant heterointerfaces and oxygen vacancies that enhance the catalytic activity [19,20,21,22,23]. The unique electronic interactions between Mo and other metal components in the alloy result in a more favorable distribution of d-electron density, further boosting the electrocatalytic efficiency [24,25,26].

Recent studies have shown that Ni-Co-Mo alloys achieved lower overpotentials and higher current densities than traditional noble metal electrocatalysts like Pt and Ir, making them promising alternatives for large-scale applications [16,27]. Mo also improves the corrosion resistance, critical for the long-term stability in alkaline environments [28]. The synergistic effects of Mo with other alloying elements such as iron have also been explored to further enhance the OER performance of Ni-Co-based electrocatalysts [29].

Beyond the metal alloy modifications with Mo, incorporating boron (B), a metalloid, into Ni-Co-Mo alloys is gaining interest due to its ability to enhance the electrocatalytic efficiency. Boron can effectively alter the electronic density of states of host metals by donating electrons from its 2p-orbitals to the d-orbitals of transition metals, facilitating proton–electron coupling and improving the OER kinetics [30]. Metal boride electrocatalysts often undergo in situ oxidation during OER, forming a core–shell structure (MB@MO_x_) with metal boride cores and oxyhydroxide shells. This restructuring may result in the leaching of B, which can be beneficial or detrimental depending on the system. The leaching of B from the electrocatalyst surface can increase the ECSA and OER rate, although it may also lead to deactivation in some binary systems. The extent of B retention or leaching depends on the alloy’s composition and structure—amorphous ternary/quaternary systems like Fe–Co–Ni–B retain B better than binary ones like Ni–B, where B leaches as BO_2_^−^ promoting surface reconstruction into active γ-NiOOH [31,32,33]. However, in binary systems like amorphous NiB, boron dissolution occurs as BO_2_^−^ during the OER, promoting surface reconstruction into active γ-NiOOH and lowering the energy barrier of the rate-determining step (*O → *OOH) [33]. Similarly, boron-doped Co–Ni oxides exhibit minimal leaching (ppb levels) under alkaline conditions, maintaining structural integrity due to the multi-metal synergy [13]. In contrast, boron in Pt–Ni/Ni–B composites oxidizes to BO^δ+^ species but remains anchored within the amorphous membrane, enhancing the corrosion resistance via electron transfer effects [34]. These findings highlight that multi-component alloys mitigate boron leaching through synergistic stabilization, while simpler systems leverage controlled dissolution for activity enhancement. The precise mechanism of B’s catalytic role in the OER is not yet fully understood, but the collective findings suggest that it contributes positively to the OER activity [30,35].

The synergistic effects of Mo and B in Ni-Co alloys are expected to yield materials with a superior OER activity and stability. However, despite their promise, the specific roles of Mo and B in Ni-Co-Mo-B alloys remain underexplored. This work aims to synthesize and evaluate a series of NiCoMoB-based electrocatalysts with varying Mo/B ratios using cryogenic ball milling—a precise, scalable technique that addresses the challenges in traditional chemical synthesis. Boron, being light with a small atomic radius, is difficult to incorporate via conventional methods such as co-precipitation, sol–gel, or impregnation, as it tends to form volatile compounds or leach during the processing. Cryomilling at low temperatures enables precise control over amorphous mixed-transition-metal compositions, ensuring a uniform element distribution and preventing the oxidation of sensitive metals such as Mo, which is hard to stabilize in its metallic form via chemical routes. The second objective is to systematically examine how Mo and B influence the OER activity and stability of the resulting alloys. By tuning the Mo/B ratio, we explore how structural changes, particularly the balance between the short-range chemical order and long-range structural order—affect the electrochemical performance. This configuration is expected to increase the number of catalytically active, coordinatively unsaturated sites that enhance the electrocatalytic efficiency.

Overall, this study offers a scalable and industrially viable approach to producing compositionally tunable alloy nanoparticles while also providing insights for designing high-performance electrocatalysts for energy conversion.

## 2. Results and Discussion

### 2.1. Phase Analysis and Particle Morphology

XRD was carried out to assess the degree of crystallinity and phase identification of all the samples, providing insights into the phase evolution across different alloy compositions. In Figure 1, both NiCoMo and NiCoB display similar XRD spectra with notable peak broadening. In the process of cryomilling of powder, peak broadening can be due to microstructural changes and phase transformations occurring for particles as well as induced micro-strain, and therefore, one needs to carefully deconvolute the broadening due to the crystallite size. The crystallite size of the cryomilled powder has been estimated from the peak broadening using the Debye–Scherrer (Equation (1)) approach after subtracting the broadening due to the XRD machine.D = K λ/β cos θ(1)
where:D: mean crystallite size (nm);K: shape factor (assumed to be 0.89, unitless);λ: wavelength of the X-ray used for diffraction (0.15418 nm);β: FWHM of the diffraction peak (rad, corrected for instrumental broadening);θ: Bragg angle (2θ/2) (°).

The instrumental broadening is accounted for by determining the full width at the half maximum (FWHM) of the selected peak. To obtain the corrected broadening, the instrumental contribution (β instrument) must be subtracted from the measured broadening (β measured). This is achieved by using the following equation:β = (β^2^
_measured_ − β^2^
_instrument_)^1/2^(2)
where β represents the corrected peak broadening, which is subsequently used for further microstructural analysis.

The average crystallite size for NiCoMo, NiCoB, NiCoMo_x_B_1−x_, and NiCoMo_y_B_y_ are 4.43 nm, 5.97 nm, 3.57 nm, and 3.05 nm, respectively. The variation in values among the samples may be attributed to the differences in the alloying elements ratio, and phase stability during the processing.

The XRD pattern of NiCoMo and NiCoB indicated peaks located at 44.4°, 51.8°, 76.4°, 92.9°, and 98.4° which are attributed to the (111), (200), (220), (311), and (222) planes, respectively, which align well with the FCC structure of the NiCo solid solution (ICSD 062-4443). In the samples containing boron, such as NiCoB and NiCoMoB, no discernible peaks corresponding to boron, or its compounds, were detected. This absence is likely due to the relatively low scattering power of boron compared to heavier elements such as Ni, Co, and Mo, as reported in previous literature [13,36]. The NiCoMo-containing samples also demonstrated a pattern matching MoNi_4_ (PDF 003-1036).

The NiCoMo_x_B_1−x_ and NiCoMo_y_B_y_ exhibited two extra peaks around 40.5° and 58.5° relative to NiCoMo and NiCoB, which may correspond to δ-NiCoMo (PDF 009-0267) likely synthesized via melt quenching, and it is not fully characterized or reported in the literature/references. These two peaks also closely match the Mo reference pattern (PCDS 089-5023), indicating the presence of a Mo-rich phase in the NiCoMoB samples. Based on the literature [36], the most likely boride phase that could form after prolonged milling is Ni_3_B. However, this phase was not detected in the XRD patterns of the 6 h cryomilled samples, suggesting that either the milling time was insufficient to induce its formation or that the phase formed is below the detection limits of the technique used. The absence of Ni_3_B in these samples may also suggest that boron is incorporated in a different form, such as in amorphous regions.

The magnified diffraction patterns for NiCoMo_x_B_1−x_ and NiCoMo_y_B_y_ shown in Figure 2 are similarly characteristic but show a slight shift to lower 2θ angles compared to the NiCoMo and NiCoB samples, suggesting that Mo and B are incorporated into the NiCo FCC lattice, resulting in an increase in the lattice parameter.

The microscopic morphology and EDS mapping of NiCoMo, NiCoB, NiCoMo_x_B_1−x_, and NiCoMo_y_B_y_ samples were analyzed via SEM both before and after the SA-HEBM process (Figure 3 and Appendix A). The milled powders showed irregular particle shapes with faceted surfaces due to their brittle nature at cryogenic temperatures. The microparticle alloys exhibited significant agglomeration, with average particle sizes of 10 ± 4 μm for NiCoMo and 15 ± 3 μm for NiCoB. The NiCoMo_x_B_1−x_ and NiCoMo_y_B_y_ alloys had average particle sizes of 5 ± 3 μm and 10 ± 6 μm, respectively. Despite the observed agglomeration, the particle sizes were reduced by 2–3 orders of magnitude compared to the initial elemental powders. Additionally, NiCoMo_y_B_y_-SA nanoparticles exhibited an average particle size of 50 ± 10 nm. These nanoparticles exhibited round and irregular morphologies, resembling their cryomilled powder form, suggesting similar fracturing mechanisms. In contrast, the NiCoMo_y_B_y_ nanoparticles produced via SA-HEBM had smoother surfaces compared to the cryomilled powder, likely due to the reduced agglomeration during milling in the presence of a surfactant.

The EDS technique enables the visualization of the spatial distribution of the constituent metallic elements with a high resolution. The elemental mapping for NiCoMo and NiCoMo_y_B_y_ shown in Figure 3a,b and Appendix A demonstrates successful alloying, evidenced by the homogeneous distribution of each element across the sample surface. The observed uniform dispersion indicates an effective alloying process, with all the elements well integrated within the material matrix. Additionally, the mapping highlights any regions of elemental segregation or incomplete alloying, offering insights into the efficiency of the cryomilling synthesis method. This elemental homogeneity is essential for achieving consistent material properties across the alloy. Notably, the distribution of elemental oxygen largely coincides with both the Mo and B patterns, suggesting the localized oxidation of Mo upon air exposure. Figure 3c,d, also depict the morphology of NiCoB and NiCoMo_x_B_1−x_ with the elemental mapping results presented in Appendix A. Figure 3d displays NiCoMo_y_B_y_ nanoparticles with particle sizes below 100 nm, indicating that the use of DDPA as a surfactant effectively facilitates the reduction in the alloyed particle size [37].

To confirm the bulk powder composition, ICP-OES analysis was performed. The ICP results and the nominal ratio of powder compositions are displayed in Table 1. The ICP analysis of the cryomilled samples shows a strong correlation between the measured atomic compositions and the intended nominal compositions, confirming the effectiveness of the cryomilling process in achieving the desired alloy ratios. The elemental analysis was conducted at wavelengths of 221.65 nm, 228.62 nm, 204.60 nm, and 182.58 nm for Ni, Co, Mo, and B, respectively. Despite the overall agreement, a minor reduction in the Mo and B content was detected in the NiCoMoB samples. This reduction is likely due to the presence of undigested trace amounts of molybdenum oxides (specifically Mo_2_O_3_) and molybdenum borides such as Mo_x_B. Mo_2_O_3_ is less soluble in acidic media compared to metallic Mo alloys, potentially leading to incomplete digestion during the ICP sample preparation. Additionally, molybdenum borides have strong covalent bonds, making them highly resistant to acid dissolution. Consequently, both Mo_2_O_3_ and MoxB may have been partially removed during the filtration, accounting for the observed reduction in the Mo and B values. This highlights the analytical challenges of fully dissolving the oxide and metal boride phases to achieve the complete elemental recovery in the acid-digested samples for the ICP analysis.

### 2.2. Electrochemical Performance

The OER activities were measured via linear sweep voltammetry (LSV, anodic sweep from 1.1 to 1.75 V vs. RHE) at 1 mV/s^1^ in 1 M KOH, and the current densities were normalized to the geometric area of the electrode (Figure 4a). The polarization curves show the current density as a function of the applied potential for different samples, with each curve representing a unique sample configuration. The OER activity of the NiCo-based alloys can be assessed by comparing the overpotentials required to achieve a certain current density, typically 10 mA/cm^2^, which is reported in Figure 4c. The overpotential for NiCoB, NiCoMo, NiCoMo_x_B_1−x_, NiCoMo_y_B_y_, and NiCoMo_y_B_y_-SA are 399, 345, 293, 284, and 270 mV, respectively, with the latter being amongst the lowest overpotentials measured compared to other NiCo-based OER electrocatalysts reported [38,39,40]. It was observed via LSV that the ternary alloys, NiCoB and NiCoMo, are the least active alloys in relation to catalyzing the OER. A higher OER activity was found with the NiCoMoB samples which showed that the simultaneous use of both Mo and B in the NiCo-based electrocatalyst plays a significant role in enhancing the overall electrocatalytic activity for the OER.

The polarization curve normalized to the ECSA of the electrodes is also shown in Appendix A. Based on the results, the overpotential at 1 mA/cm^2^ ECSA follows a similar trend: NiCoB (370 mV) < NiCoMo (340 mV) < NiCoMo_x_B_1−x_ (300 mV) < NiCoMo_y_B_y_ (280 mV) < NiCoMo_y_B_y_-SA (260 mV). A key observation is the difference in the overpotential between NiCoMo and NiCoB, which is 30 mV when LSV is normalized via the ECSA but increases to 54 mV when normalized via the geometric area. This discrepancy highlights the importance of the normalization methods in evaluating the electrocatalytic performance. The ECSA normalization reveals the intrinsic activity per active site, eliminating the extrinsic influences such as the electrode loading and morphology. The lower overpotential observed via the ECSA-normalized LSV for NiCoB suggests that its active sites are inherently more efficient in facilitating the OER compared to the geometric-area-normalized results. This difference may point to the critical role of boron in enhancing the intrinsic OER activity, particularly in NiCoMoB alloys, where the variations in the Mo/B ratio influence the catalytic performance. These findings suggest that boron incorporation optimized the active site utilization, contributing to the improved electrocatalytic efficiency.

The sample with a Mo/B ratio of 1 (NiCoMo_y_B_y_) demonstrates a higher OER activity than the sample with a Mo/B ratio of 0.49 (NiCoMo_x_B_1−x_). This optimal ratio likely balances the electronic and structural contributions from Mo and B, maximizing the density and activity of the available electrocatalytic sites. The addition of boron led to higher current densities as well as lower Tafel slopes in agreement with the work reported by [13,41,42].

For the NiCoMo_y_B_y_ sample synthesized in a nanoparticle form via surfactant-assistant high-energy ball milling (NiCoMo_y_B_y_-SA), the OER activity further increases. This enhanced performance can be attributed to the higher active surface area provided by the nanoparticles, which expose more electrocatalytic sites and allow for more effective electron transfer [43]. The nanoparticle morphology also improves the mass transport, facilitating the easier diffusion of the reactants and products to and from the active sites. This structural refinement, combined with the optimal Mo/B composition, resulted in an electrocatalyst that exhibited lower overpotential requirements and higher current densities, underscoring its potential as a highly efficient OER electrocatalyst.

In the NiCoMo_y_B_y_-SA electrocatalyst, the observed electrocatalytic activity is governed by two competing effects. The agglomeration of nanopowders tends to reduce the electrocatalytic performance by limiting the active surface area and hindering the mass transport, effectively lowering the accessibility of the active sites. Conversely, the intrinsic activity, optimized through the compositional design, enhances the electrocatalytic performance by facilitating higher current densities at lower overpotentials. This interplay between nanoparticle agglomeration, which detracts from the activity, and the inherent electrocatalytic efficacy of the alloy composition is central to the electrochemical behavior observed in the LSV analysis.

It is worth noting that an oxidation peak which appeared in the LSV curves of the NiCoMoB catalysts prior to the OER was attributed to the Ni^2+^(OH)_2_ → Ni^3+δ^OOH transition [44]. The Tafel slopes are calculated by using Equations (3) and (4), where α is the charge transfer coefficient, b is the Tafel slope, and i is the current density.*η* = *a* + *b* log (i)(3)α = 2.303 *R T*/*b F*(4)

The NiCoMo_x_B_1−x_ and NiCoMo_y_B_y_, having Tafel slopes of 42 and 38 mV/dec, exhibited comparable values suggesting that the reaction mechanism remains unchanged with variations in the Mo/B ratio (Figure 4b). This indicates that the OER mechanism is more likely governed by the constant Ni/Co ratio in both samples, rather than the Mo/B content. Notably, although NiCoMo_y_B_y_-SA demonstrates a superior OER activity compared to its bulk powder counterpart, the reaction mechanism appears to be consistent with that of the microstructured particles, as evidenced by its similar Tafel slope. This suggests that despite the enhanced activity, the underlying electrochemical behavior remains unchanged.

The onset potential of an electrocatalyst for the OER is a critical parameter that determines the energy required to initiate the electrochemical reaction. A lower onset potential indicates that the electrocatalyst is more efficient in promoting the OER, as it requires less voltage to overcome the activation energy barrier. The NiCoB alloy, with an onset potential of 1.56 V, showed the highest onset potential, suggesting a lower activity and overpotential at E_onset_ of 335 mV (Figure 4c) which means less favorable kinetics for the OER. NiCoMo and NiCoMo_x_B_1−x_ alloys fall in between, with onset potentials of 1.52 and 1.49 V, and the overpotentials at this point are 291 and 266 mV, respectively, reflecting a moderate OER activity. The onset potential and the overpotential at this point for NiCoMo_y_B_y_ and NiCoMo_y_B_y_-SA are 1.48, 1.45, 256, and 229 mV, respectively, which shows a clear trend in the electrocatalytic activity of these materials. The NiCoMo_y_B_y_-SA alloy demonstrates the lowest onset potential, indicating that it requires the least energy to initiate the OER, suggesting the superior electrocatalytic efficiency of nanoparticles compared to the other samples. EIS was employed to gain deeper insights into the reaction kinetics, highlighting the critical role of structural modifications in enhancing the OER activity of NiCoMo_y_B_y_.

Figure 4d presents the Nyquist plot for NiCoMo_y_B_y_ microparticles and NiCoMo_y_B_y_-SA nanoparticles. Electrochemical impedance spectroscopy (EIS) was conducted at 1.5 V and 1.6 V vs. RHE to study the oxygen evolution reaction kinetics of the synthesized electrocatalysts. The potential of 1.5 V vs. RHE was chosen as it is near the onset potential for the OER, allowing the investigation of the initial charge transfer resistance (R_ct_). The potential of 1.6 V vs. RHE lies within the potential range at which the OER proceeds at a significant rate, providing insights into the electrocatalyst’s performance under industrial operating conditions. This potential enables a comprehensive understanding of the electrocatalyst’s behavior at both early and more advanced stages of the OER.

The two semicircles in the Nyquist plots can be ascribed to the charge transfer on the electrodes, as both semicircles decrease in size with increasing potential, indicating two separate electrochemical processes, each with its own time constant. The larger semicircle at lower frequencies usually corresponds to the mass transfer or ion diffusion at the electrode–electrolyte interface, while the smaller semicircle at higher frequencies represents the R_ct_ associated with the electron transfer between the electrode and electrolyte.

The Nyquist plots reveal a distinct difference in the charge transfer resistance (R_ct_) between the NiCoMo_y_B_y_ microparticles and NiCoMo_y_B_y_-SA nanoparticles. At 1.5 V vs. RHE, the R_ct_ of NiCoMoyBy microparticles was approximately 233.7 Ω, whereas NiCoMo_y_B_y_-SA nanoparticles showed a significantly lower R_ct_ of 49.9 Ω. At 1.6 V vs. RHE, these values further decreased to 45.2 Ω for NiCoMo_y_B_y_ and 30.9 Ω for NiCoMo_y_B_y_-SA. NiCoMo_y_B_y_-SA exhibited the lowest R_ct_ at both potentials, indicating a more efficient charge transfer at the electrode–electrolyte interface. This lower R_ct_ suggests that NiCoMo_y_B_y_-SA has superior electrochemical kinetics, facilitating faster electron transport during the OER. In contrast, the NiCoMo_y_B_y_ sample displays a significantly larger R_ct_, implying a slower charge transfer and potentially less accessible active sites for the OER. This disparity in R_ct_ values highlights the enhanced electrocatalytic performance of the NiCoMo_y_B_y_-SA structure, which may be due to the improved surface area, optimized electronic structure, or a more favorable active site distribution in the sample. The Nyquist plots of NiCoMo_y_B_y_-SA are clearly composed of two semicircles.

Figure 5 presents the chronoamperometry profiles for the sample, NiCoMo_y_B_y_-SA, conducted at a current density of 10 mA/cm^2^ over 8 days, followed by 50 mA/cm^2^ for an additional 2 days, and a final stage at 100 mA/cm^2^ sustained for 10 days in 1 M pre-electrolyzed KOH at 30 °C. The potential required to achieve a current density of 10 mA/cm^2^ decreased in the first few hours indicating electrocatalyst activation. This potential subsequently stabilizes, maintaining a steady overpotential for 8 days with a minimal increase. Comparable stability (0.02 mV/h) is observed at 50 mA/cm^2^. After 10 days, chronoamperometry tests at a higher current density of 100 mA/cm^2^ showed only a slight decline in the electrocatalytic activity, affirming the high stability of the electrocatalyst. The element leaching during the OER is a critical factor that directly impacts the stability and efficiency of electrocatalysts. This process involves the migration of metal ions from the electrocatalyst into the electrolyte, often resulting in structural degradation and modified electrochemical behavior. To evaluate the stability and leaching behavior of the OER electrocatalyst under harsh operational conditions, chronoamperometry was performed at a high current density of 500 mA/cm^2^, and the electrolyte was subsequently analyzed using inductively coupled plasma (ICP) spectroscopy. While the electrocatalyst’s activity was assessed at a practical current density of 100 mA/cm^2^, the higher current density of 500 mA/cm^2^ was employed to accelerate any potential degradation processes, such as element leaching, within a shorter time period. This approach simulates extreme operating conditions, providing critical insights into the electrocatalyst’s stability and the extent of element leaching under highly demanding environments, which is essential for assessing its long-term applicability in real-world OER systems. Leaching, driven by the concentration gradients between the electrocatalyst surface and the alkaline electrolyte, induces structural changes that can alter the electrocatalytic performance. ICP-OES analysis was conducted on the electrolyte solution at intervals of 0, 0.5, 1, 3, 6, 9, 12, 24, 30, 50, and 100 h during the stability testing. The analysis of the fresh electrolyte prior to the OER confirms the absence of detectable traces of nickel (Ni), cobalt (Co), molybdenum (Mo), and boron (B). However, after 30 min of chronoamperometry testing, a minimal concentration of Mo is observed in the electrolyte, indicating the onset of elemental leaching.

Upon extending the chronoamperometry test to 1 h, more pronounced leaching behavior is evident. Trace amounts of Ni and Co are detected in the electrolyte, albeit at negligible concentrations. In contrast, a significantly higher concentration of boron is observed, suggesting the substantial dissolution of boron from the NiCoMoB electrocatalyst during the OER process. This differential leaching behavior is further supported by the elemental concentration data over time, as depicted in Figure 5b. The graph shows a gradual increase in the boron concentration, reaching a peak at longer durations, while the concentrations of Ni and Co remain relatively low throughout the testing period.

Following the structural and electrochemical analyses, XPS was utilized to investigate the metal oxidation state changes and the chemical environment of oxygen species, providing insights into the chemical and structural modifications induced by anodic cycling. XPS measurements were conducted on the alloy nanoparticles both before and after 50 anodic cycles. The XPS survey of NiCoMo_y_B_y_-SA nanoparticles is presented in Figure 6a. The XPS survey verifies the presence of Co, Ni, Mo, Co, B, O, and C species which is consistent with the EDS analysis. The high-resolution XPS spectra of Ni 2p, Co 2p, Mo 3d, B 1s, and O 1s are also presented in Figure 6.

Figure 6b exhibits features corresponding to the multiple nickel oxidation states. The primary peak at ~855.5 eV (Ni 2p_3/2_) and its associated spin–orbit satellite at ~873.2 eV were assigned to Ni^2+^, indicative of the presence of NiO and/or Ni(OH)_2_ species [45,46]. The presence of Ni^2+^ is further confirmed by the shake-up satellite feature at ~861.4 eV and ~879.4. A peak at ~853.4 eV (Ni 2p_3/2_) and its spin–orbit pair at ~870.8 eV corresponds to Ni^0^, indicating the presence of metallic nickel on the surface [47,48]. A small component at ~856.9 eV (Ni 2p_3_/_2_) was also observed, corresponding to Ni^3+^, potentially indicative of a small amount of γ-NiOOH [49]. Following the OER polarization post polarization, significant changes were observed in the Ni 2p spectra (Figure 6b). The peak intensities associated with metallic nickel (Ni^0^) at ~853.4 eV and ~870.8 eV were reduced, suggesting the surface oxidation of Ni^0^ during the OER process. The reduction in metallic Ni signal suggests that metallic nickel is likely oxidized to form Ni^2+^ or Ni^3+^ species during the OER process. The Ni^2+^ peaks at ~855.5 eV and ~873.2 eV remain present, indicating the stability of the NiO/Ni(OH)_2_ reservoir even after electrochemical oxidation [48,49]. Clearly, the peak intensity of the Ni^3+^ component at ~856.9 eV increased substantially after the OER polarization. The shake-up satellite peaks at ~861.4 eV and ~879.4 eV remained consistent, confirming the presence of Ni^2+^ species. The persistent presence of NiO/Ni(OH)_2_ acts as a reservoir for Ni^3+^ formation, ensuring a continuous supply of the active NiOOH species during prolonged OER operation. These findings highlight the crucial role of in situ surface oxidation and the formation of NiOOH in promoting the OER activity on nickel-based electrocatalysts. As shown in Figure 6c, the Co 2p spectrum of the NiCoMo_y_B_y_-SA electrocatalyst can be resolved fairly well with three spin–orbit satellites from three chemically different Co entities (Co^0^, CoO, and Co (OH)_2_) and Ni 2p shake-up satellites. The Co 2p XPS analysis reveals significant changes in the cobalt oxidation states following the OER polarization. Before the OER, the presence of Co^2+^ (CoO/Co(OH)_2_) at ~779.9 eV and Co^3+^ (CoOOH) at ~781.2 eV indicates a mixed-valence state, with a satellite feature at ~784.6 eV characteristic of Co^2+^. After the OER, the Co^2+^ peak area is reduced, suggesting partial oxidation, while the Co^3+^ peak increases, confirming the formation of the OER-active Co-doped NiOOH. The enhanced shake-up satellite at ~796.9 eV further supports stronger Co^3+^ contributions post the OER. These results confirm the dynamic oxidation of Co species, highlighting the Co^2+^ to Co^3+^ transition as a key factor in the OER activity.

The Mo 3d spectra (Figure 6e) show the deconvoluted Mo peaks, and the splitting width of 3.1 eV was ascribed to Mo 3d_5/2_ and Mo 3d_3/2_, respectively, characteristic of the hexavalent Mo^6+^ oxidation state [50,51]. The XPS analysis of the cryomilled Ni-Co-Mo-B system reveals complex bonding states, including boride formation, oxidation, and structural disorder. The Mo 3d peaks at 229, 232.1, 229.2, 232.3, 231.2, 234.3, 233.4, and 236.5 eV indicate Mo-B bonding, and various oxidation states (Mo^4+^/Mo^5+^/Mo^6+^). Cryomilling facilitates Mo-B alloying (229 eV), while oxidation leads to MoO_2_ (229.2 eV), Mo^5+^ (231.2 eV), and MoO_3_ (233.4 eV). Following the OER polarization, the 229 and 232.1 eV peaks disappear entirely, suggesting that the corresponding Mo species are transformed into other oxidation states. Simultaneously, the remaining peaks shift to lower binding energies (e.g., 229.2 eV to 228 eV, 232.3 eV to 231.1 eV), while exhibiting increased FWHM values, indicative of the greater disorder and a wider distribution of Mo environments. This suggests a dynamic surface restructuring process during the OER, where specific Mo species are selectively changed, and the remaining molybdenum undergoes reduction and amorphization.

The electrochemical polarization induced significant alterations in the surface chemistry of the Ni-Co-Mo-B catalyst, as evidenced by the XPS analysis of the B 1s region. Prior to polarization, the B 1s spectrum exhibited four distinct chemical states (187.84, 189.54, 190.71, and 192 eV), reflecting the presence of metal borides such as Mo-B [52,53], partially oxidized boron, and B_2_O_3_ species. Post-polarization, the disappearance of the metal boride peak (187.84 eV) and the emergence of higher binding energy features (191.15–192.5 eV) confirmed the preferential oxidation of metal–boron bonds and the formation of a boron oxide-rich surface layer. Notably, while the overall B 1s intensity remained relatively constant, a decrease in the peak FWHM was observed after the polarization, indicating a more homogeneous distribution of the boron oxidation states and suggesting a surface restructuring process that may enhance the catalyst stability through passivation. The observed chemical redistribution and peak narrowing highlight the dynamic nature of the catalyst surface under electrochemical conditions, providing critical insights into the design of boron-containing electrocatalysts with improved performance.

O 1s XPS spectra of the Ni-Co-Mo-B catalyst revealed surface oxygen evolution during the OER polarization in KOH (Figure 6f). Prior to the OER, the O 1s spectrum deconvolution showed three components: lattice oxygen (529.35 eV), hydroxyl/oxygen vacancies (530.21 eV), and adsorbed water (531.42 eV). Post the OER, the lattice oxygen shifts to 529.86 eV, possibly due to the increased metal oxidation [54,55]. This shift reflects three interrelated processes: (1) electron density loss from oxygen due to the lattice oxygen extraction during the OER (2OH^−^ → O_2_ + 2H^+^ + 4e^−^) [56], creating electron-deficient sites; (2) surface reconstruction into higher-valent oxyhydroxides (e.g., NiOOH), where the increased metal oxidation states (Ni^3+^) withdraw the electron density from the bonded oxygen [57]; and (3) the influence of adsorbed transient intermediates (*OH, *OOH), further deshelling the oxygen nuclei by localizing the positive charge. This BE shift is mechanistically linked to the increasing effective nuclear charge on oxygen [58], as electronegative high-valence metals (e.g., Ni^3+^, d^7^ configuration) polarize M–O bonds. Notably, the magnitude of the O 1s BE shift correlated positively with the OER activity, with the catalysts exhibiting larger shifts demonstrating the lower overpotentials for O_2_ evolution. Additionally, after the polarization, the intensity of the 531.0 eV peak increases, while that of the 531.8 eV peak decreases, suggesting a transformation of adsorbed water to the surface hydroxyls. The FWHM values also decreased, indicating more homogeneity.

This suggests the creation of active sites via surface hydroxylation. Hydroxyl groups are crucial in the OER mechanism, facilitating proton transfer and oxygen evolution. This transformation aligns with reports that surface hydroxylation is a prerequisite for an efficient OER activity, where M-OH species are the key intermediates in the formation of O-O bonds. The shift of the O 1s peaks and the increase in intensity of the hydroxyl species suggest a dynamic surface restructuring during the OER, consistent with the observed Mo 3d changes. The post-OER O 1s line shape thus serves as a fingerprint of in situ surface oxidation and intermediate accumulation, providing key insights into the catalytic activity and stability. The fitting parameters for the NiCoMo_y_B_y_-SA electrocatalyst before and after the electrochemical testing are provided in Appendix A.

## 3. Materials and Methods

### 3.1. Synthesis of Ni-Co-Based Electrocatalysts

This study aimed to investigate the individual and synergistic effects of Mo and B on the electrocatalytic performance of Ni-Co-Mo-B alloys, while maintaining a constant Ni/Co ratio to isolate the contributions of these additional elements. Maintaining a fixed Ni/Co ratio is critical, as it allows for the direct comparison of the effects of Mo and B without interference from the variations in the host metal content, ensuring that the observed changes in the electrocatalytic performance are attributable to the presence of these alloying elements. To achieve this, NiCoMoB electrodes were systematically designed and investigated with respect to Mo and B concentrations. Cryomilling and post-milling processes assisted by using surfactants were used as the alloy preparation methods and are detailed below.

The synthesis of four alloys, designated as NiCoMo, NiCoB, NiCoMo_x_B_1−X_ (x < 1 − x), and NiCoMo_y_B_y_ in the form of micron-sized particles, was achieved through high-energy cryomilling following the procedure previously reported by Cole et al. [59] and Ghobrial [60]. The Ni/Co atomic ratio was fixed across all the samples. The synthesis procedure is depicted as a graphical abstract in Figure 7. Micron-sized NiCoMo alloy powders were initially synthesized via the cryogenic mechanical alloying of elemental Ni, Co, and Mo powders at liquid nitrogen temperatures, ca −196 °C, resulting in a disordered structure. This process was carried out using a Retsch CryoMill using 5 mL stainless steel milling vials. Each vial was loaded with elemental powders in an Ar-filled glove box along with two stainless steel balls (7 mm diameter, 1.4 g each) to achieve a ball-to-powder ratio (BPR) of 10:1. The masses of Ni, Co, and Mo powders were calculated based on the weight of the milling balls to ensure the desired BPR. The milling was performed at a frequency of 30 Hz for 6 h under sustained cryogenic conditions, maintained by using liquid nitrogen circulating within the cooling jacket. NiCoB, NiCoMo, NiCoMo_x_B_1−x_, and NiCoMo_y_B_y_ were synthesized using the same method.

After cryomilling, surfactant-assisted high-energy ball milling (SA-HEBM) was used to reduce the particle size of the NiCoMo_y_B_y_ powder to the nanoscale range labeled as NiCoMo_y_B_y_-SA. The surfactant employed in this step was dodecylphosphonic acid (DDPA) (95%) (20 wt.% of the powder mass), and the milling medium was anhydrous ethanol (100 wt.% of the powder mass). The surfactant and ethanol were added simultaneously to stainless steel vials containing the previously milled NiCoMo_y_B_y_ powder and milling balls. SA-HEBM was run at 30 Hz with a fixed BPR of 50:1 for 10 h at room temperature. To avoid overheating, a cycling method of 30 min on and 82 min off was employed.

Following SA-HEBM, the nanoparticles were washed and then extracted as described below to prepare them for the electrochemical testing and characterization using sonication and centrifugation. Initially, the particles were suspended in scintillation vials containing 20 mL of 95% anhydrous ethanol and bath-sonicated for 60 min. Following that, the suspension was transferred to 50 mL centrifuge tubes and centrifuged at a low speed (150 rpm) for 40 min to separate the bigger sub-micron particles (>300 nm) from the nanoparticles. The residual nanoparticle suspension was transferred to new 50 mL centrifuge tubes and centrifuged at 7800 rpm for 50 min to separate the nanoparticles. The supernatant was pipetted out, and fresh ethanol was used to resuspend the nanoparticles followed by another 60 min of sonication. To ensure that all the surfactants and larger particles were removed, the washing process was repeated multiple times until the extracted liquid was clear. The cleaned nanoparticles were dried at 60 °C in an oven overnight. The dried nanoparticles were then transferred to and stored in a 2 mL vial for future use.

### 3.2. Physicochemical Characterization

The elemental compositions of the powders were tracked after milling using inductively coupled plasma atomic emission spectroscopy (ICP-OES, Thermo Scientific iCAP Pro, Thermo Fisher Scientific Inc., Waltham, MA, USA) to verify that there were no noticeable deviations in the elemental concentrations from the nominal values or contamination from the milling apparatus. Ni, Co, Mo, and B were analyzed using multielement standards (high-purity multi-element standard solution, QC4). For calibration, the standard solutions were prepared in concentrations of 1, 5, 10, and 20 ppm. The Rigaku MiniFlex 600 (Rigaku Corporation, Tokyo, Japan) was used to obtain X-ray diffraction patterns with Cu Kα radiation (λ = 1.5418 Å). A “step” mode of acquisition was used for the X-ray diffraction settings. This was performed over a range of 2θ of 20° to 100° with a step size of 0.02° and a speed of 2 s/step. Rigaku PDXL2 version 2.0 software was used for the analysis and peak identification after the XRD measurements. 

A high-resolution scanning electron microscope (Hitachi SU7500, Hitachi High-Tech Corporation, Tokyo, Japan) equipped with an energy-dispersive X-ray spectroscopy (EDS) detector was used to examine the produced particle sizes and morphologies. The elemental composition and molecular species of the alloy surface were analyzed using time-of-flight secondary ion mass spectrometry (ToF-SIMS, Thermo Fisher Scientific, Waltham, MA, USA) with a mass resolution of m/Δm > 10,000 and detection limits in the ppm to ppb range. Monochromatic Al Kα was used in X-ray photoelectron spectroscopy (XPS, Thermo Fisher Scientific, Waltham, MA, USA) investigations utilizing a Thermo Fisher Scientific EscaLab Xi. Ag foil was used to calibrate the XPS analyzer, and C 1s = 284.8 eV was used as the reference for the binding energy scale. Anodic cycling for the XPS measurements was conducted within an argon-filled glove box connected to the XPS vacuum transfer chamber, enabling rapid surface species analysis during cycling while preventing the exposure to airborne contaminants.

### 3.3. Electrochemical Measurements

All the electrochemical data measurements were collected using a Bio-Logic VSP-300 multichannel (Bio-Logic Science Instruments, Fontaine, France) potentiostat/galvanostat. Measurements were also conducted using a static three-electrode setup that included a fine Pt mesh counter electrode (CE) and a 1 M Hg/HgO reference electrode (RE). For all the GCE electrode tests, a 100 mL Teflon cell was filled with 100 mL of fresh Fe-free 1 M KOH (pre-electrolyzed for 72 h to minimize Fe impurities). The Hg/HgO reference electrode was calibrated using a saturated calomel electrode. The voltage was reported using the equation, E_RHE_ = E_SHE_ + 0.241 V + 0.059 × pH.

Electrocatalyst micro-inks were synthesized by mixing 8 mg of microparticles with 10 μL of 5 wt.% Nafion (Ion Power, New Castle, DE, USA) and 200 μL of isopropyl alcohol. For the nano-ink preparation, a combination of 4 mg of nanoparticles, 80 μL of 5 wt.% Nafion (Ion Power), 1 mL of deionized water, and 0.25 mL of anhydrous ethanol was employed. The working electrodes were fabricated via drop casting 0.21 mg cm^−2^ of electrocatalyst inks onto 3 mm diameter glassy carbon electrodes, which were polished using 0.05 μm colloidal silica (CH-Instruments, Bee Cave, TX, USA). These electrodes were spin dried at 400 rpm in ambient air for at least 30 min. For the steady-state chronoamperometry, working electrodes were prepared by spraying 0.21 mg cm^−2^ of electrocatalyst ink onto Ni foam. Each electrochemical test was carried out at 30 °C in a 1 M Fe-free KOH electrolyte. Before the electrochemical testing, the electrolyte was pre-electrolyzed for 72 h at −1.7 V (Hg/HgO) using an Ivium potentiostat to eliminate any trace iron contaminants. Platinum mesh electrodes used in the pre-electrolysis were periodically acid washed to remove contaminants. Using steady-state potentiostatic polarization, the potential was stepped in 20 mV increments between 0.3 and 0.8 V (Hg/HgO) then held for 10 min to quantify the Tafel slope. Linear sweep voltammetry was used to measure the anodic polarization at a scan rate of 1 mV/s^1^. Over a 20-day period, steady-state chronoamperometry was performed at 10, 50, and 100 mA/cm^2^ to assess the long-term stability.

## 4. Conclusions

In this study, we systematically investigated the role of molybdenum (Mo) and boron (B) in NiCo-based alloys for the OER. Quaternary NiCoMoB electrocatalysts demonstrated significantly enhanced OER performance compared to their ternary NiCoMo and NiCoB counterparts, with the optimal Mo/B ratio of 1 (NiCoMo_y_B_y_) exhibiting superior electrocatalytic activity due to the synergistic effects of Mo and B, which balance the electronic and structural contributions, maximizing the density and activity of the electrocatalytic sites. The nanoparticle version, NiCoMo_y_B_y_-SA, synthesized via a surfactant-assisted method, achieved the lowest overpotential (270 mV at 10 mA/cm^2^) and onset potential (1.45 V), reflecting its superior efficiency. Stability tests revealed the exceptional stability, with the electrocatalyst activating within hours at 10 mA/cm^2^, stabilizing over 7 days, and showing minimal overpotential increases (0.02 mV/h at 50 mA/cm^2^) even after 10 days at 100 mA/cm^2^. These findings highlight the potential of NiCoMoB electrocatalysts as cost-effective, stable, and highly efficient OER electrocatalysts, advancing their application in sustainable energy technologies.

## Figures and Tables

**Figure 1 molecules-30-01971-f001:**
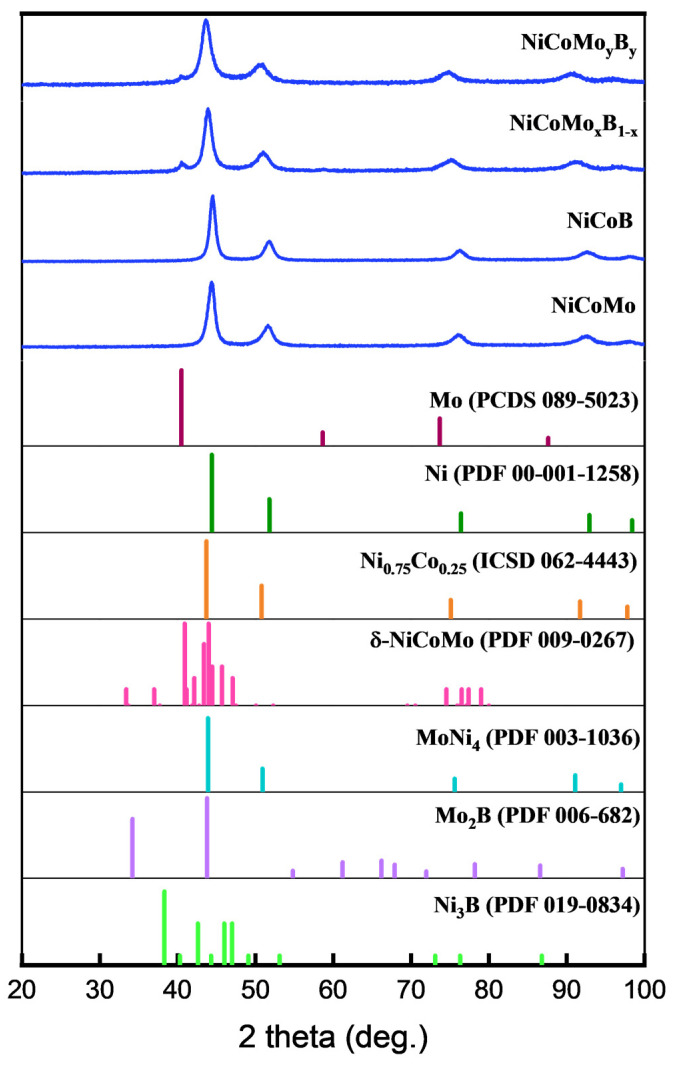
XRD spectrum of NiCoMo, NiCoB, NiCoMo_x_B_1−x_, and NiCoMoyBy, along with corresponding XRD standard spectra of related phases.

**Figure 2 molecules-30-01971-f002:**
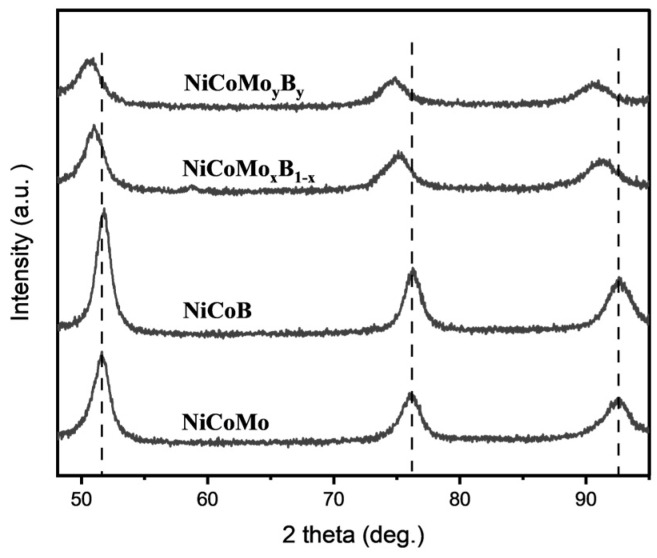
XRD spectrum of NiCoMo, NiCoB, NiCoMoxB1_x, and NiCoMoyBy on a magnified scale.

**Figure 3 molecules-30-01971-f003:**
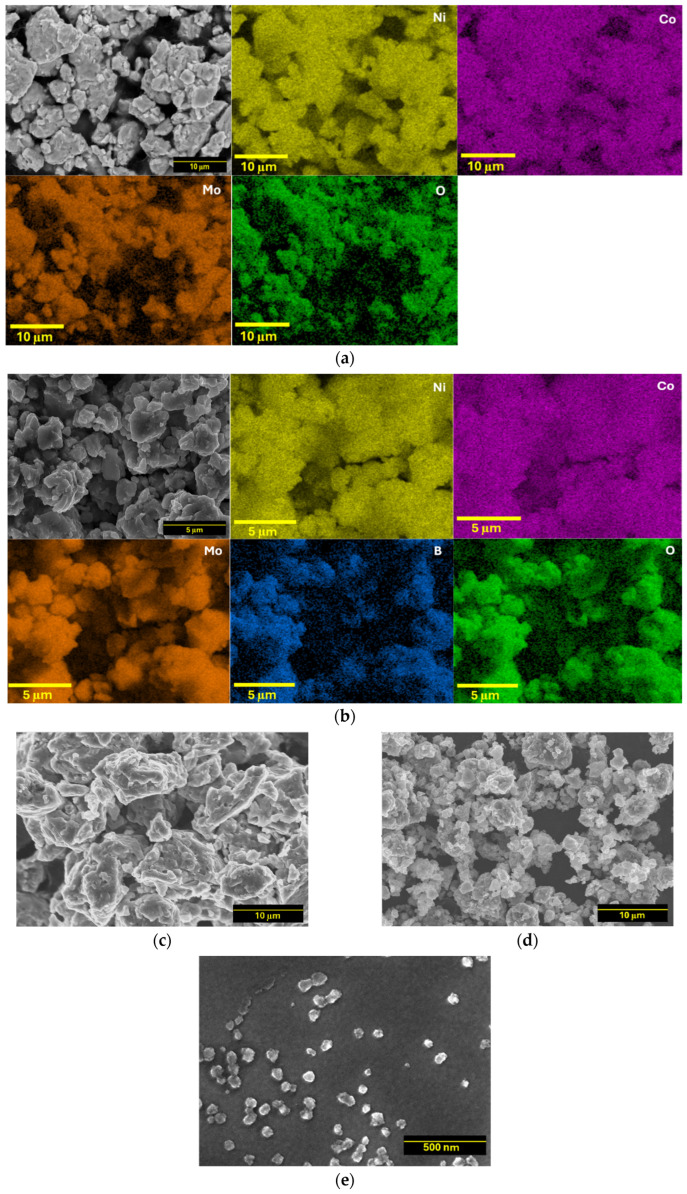
SEM images and EDS mapping of as-cryomilled (**a**) NiCoMo, (**b**) NiCoMo_y_B_y_, and SEM images of (**c**) NiCoB, (**d**) NiCoMo_x_B_1−x_, and (**e**) NiCoMo_y_B_y_-SA nanoparticles obtained after SA-HEBM.

**Figure 4 molecules-30-01971-f004:**
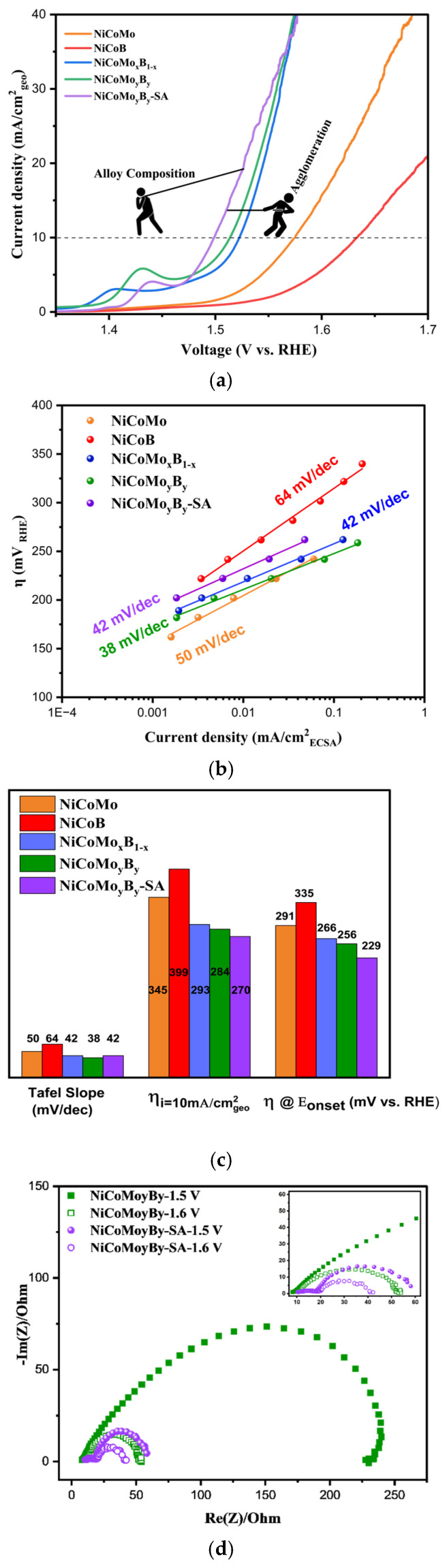
(**a**) LSV normalized by geometric area, (**b**) Tafel plots, (**c**) comparison of the corresponding Tafel slope, η at i = 10 mA/cm^2^, and onset potential of NiCoMo, NiCoB, NiCoMo_x_B_1_x_, NiCoMo_y_B_y_, and NiCoMo_y_B_y_-SA; (**d**) Nyquist plots of NiCoMo_y_B_y_ and NiCoMo_y_B_y_-SA at 1.5 and 1.6 V.

**Figure 5 molecules-30-01971-f005:**
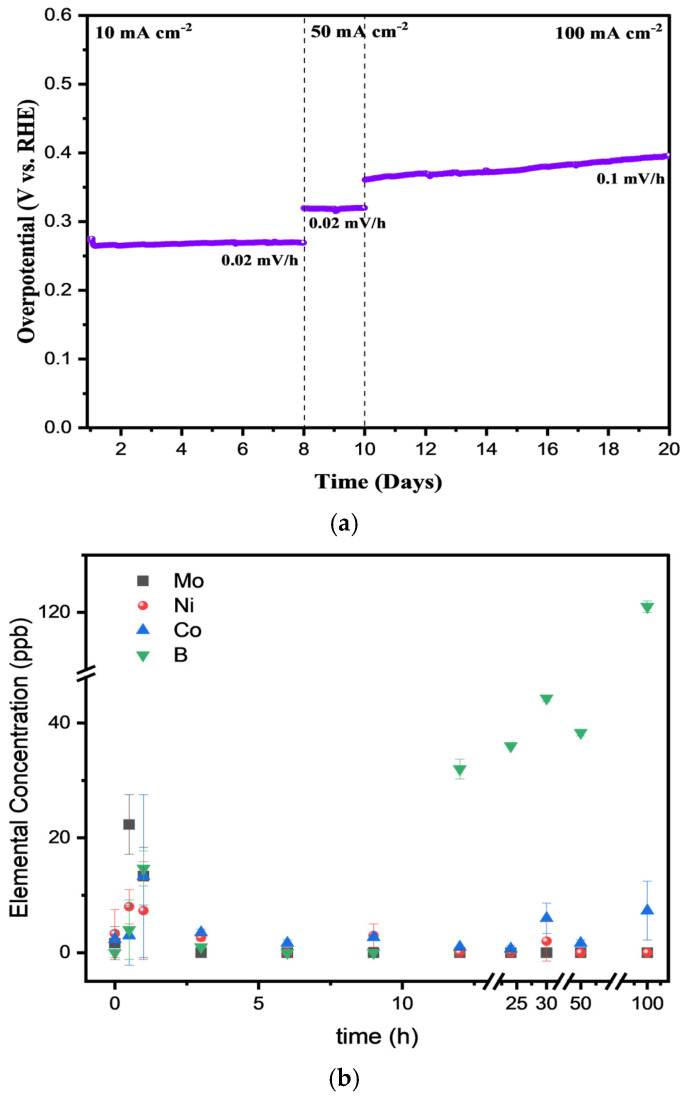
(**a**) Chronoamperometric measurements were conducted over 16 days for NiCoMo_y_B_y_-SA nanoparticles deposited on Ni foam at current densities of 10, 50, and 100 mA/cm^2^. Experiment is carried out in 1 M KOH at 30 °C, with an electrode loading of 0.21 mg cm^−2^, (**b**) ICP of electrolyte collected at the different chronoamperometry times.

**Figure 6 molecules-30-01971-f006:**
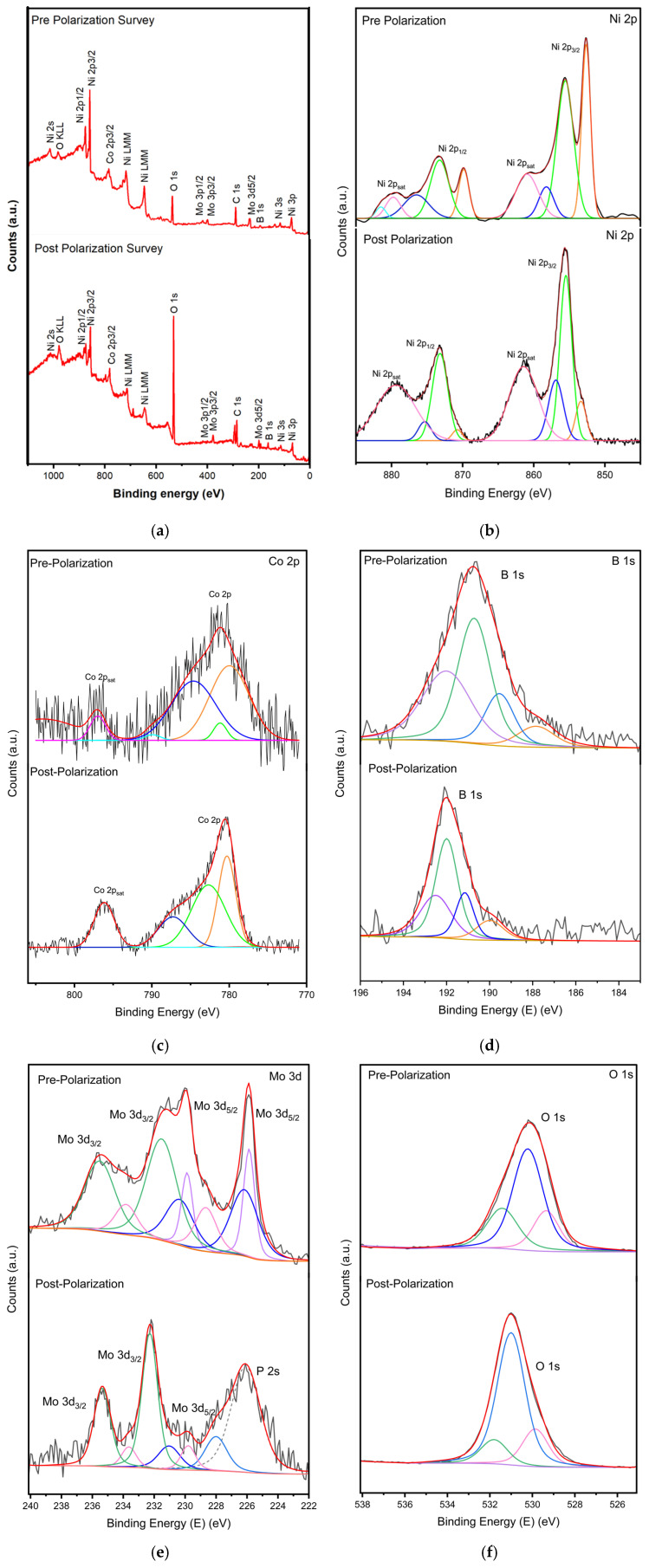
Core-level XPS spectra of (**a**) survey (**b**) Ni 2p, (**c**) Co 2p, (**d**) Mo 3d, (**e**) B 1s, and (**f**) O 1s measured for NiCoMo_y_B_y_-SA before and after OER polarization.

**Figure 7 molecules-30-01971-f007:**
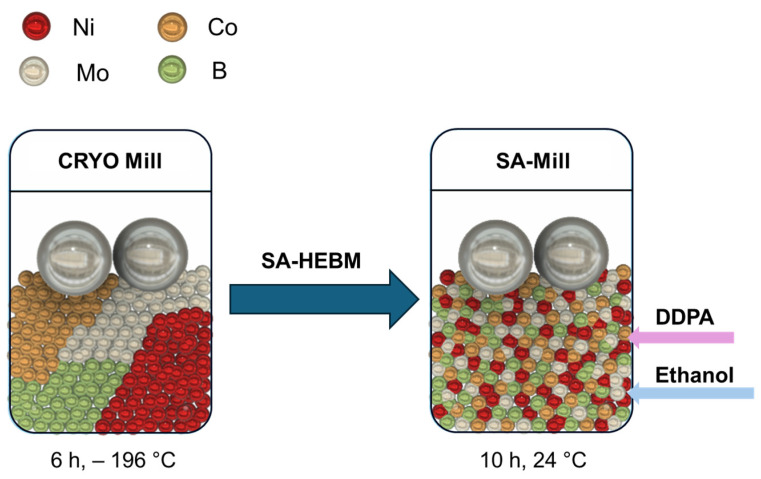
A schematic diagram of the synthesis method of NiCoMoB alloys using cryomilling and SA_HEBM.

**Table 1 molecules-30-01971-t001:** Atomic composition and elemental ratios Ni/Co and Mo/B for NiCoMo, NiCoB, NiCoMo_x_B_1−x_, and NiCoMo_y_B_y_ samples. Nominal and ICP-measured ratios are compared.

Sample	Atomic Composition				Nominal Ratio		ICP Ratio	
	Ni	Co	Mo	B	Ni/Co	Mo/B	Ni/Co	Mo/B
NiCoMo	83.09	5.81	11.09	-	15	-	14.3	-
NiCoB	72.22	5.20	-	22.58	15	-	13.9	-
NiCoMo_x_B_1−x_	68.80	4.90	7.16	19.14	15	0.49	14.0	0.40
NiCoMo_y_B_y_	72.61	5.13	10.64	11.62	15	1	14.2	0.94

## Data Availability

The original contributions presented in this study are included in the article/Appendix A.

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
