# Peer review of "Systematic Investigation of the Role of Molybdenum and Boron in NiCo-Based Alloys for the Oxygen Evolution Reaction"

_molecules, 2025, doi:10.3390/molecules30091971_

Round 1

Reviewer 1 Report

Comments and Suggestions for Authors

In this manuscript, the authors provided a systematic study on the role of molybdenum and boron in NiCo-based alloys for the oxygen evolution reaction (OER) in alkaline solutions. The optimized sample exhibited both enhanced OER activity and long-term stability. The research has good novelty and the results were clearly presented. I would like to support its publication at Molecules. However, before that, the below detailed comments are suggested to be addressed to further improve the clarity and quality of the present manuscript.

  1. Line 38 and line 81, the authors mentioned HER. First of all, HER was not defined at its first appearance. Secondly, in some of the contexts around these areas, HER might not be relevant, given that this work focuses on OER. So please make some revision to the related texts.
  2. The Introduction might appear to be a bit too long and tedious. It is suggested that the authors focus on the key context and background and the novelty of the present work and therefore make some revision to the Introduction section.
  3. Related works on water splitting can be referenced to appeal to a broader readership (e.g., doi: 10.1002/inf2.12494; 10.1016/j.matre.2024.100283).
  4. The authors designed a ratio for the synthesis of the alloys and ensured that Ni and Co had the same atomic ratio across different control samples. Did the authors check if the element ratio of the as-obtained samples was consistent with the designed ratio?
  5. In addition to the EDS mapping images, can the authors also provide the EDS spectra to corroborate the presence of each element? If possible, from EDS spectra, the authors might also be able to get the quantitative information about the element ratio.
  6. Figure 6c, the satellite peaks were designated to the Ni, but not Co. Please double check. Figure 6e, double check the labelling of Mo 3d5/2 and Mo 3d3/2; for each pair of Mo peaks, the higher intensity peak should be 3d5/2 and lower intensity one should be 3d3/2. Also, why would P peak be found in the sample NiCoMoB (which does not contain P element)?
  7. Figure 7 appears to be an image made with assistance of generative AI. Please double check and also check with journal to see the publication guidelines for this kind of figures (should it be prepared with involvement of generative AI).

Author Response

Comment 1: Line 38 and line 81, the authors mentioned HER. First of all, HER was not defined at its first appearance. Secondly, in some of the contexts around these areas, HER might not be relevant, given that this work focuses on OER. So please make some revision to the related texts.

Response 1: We thank the reviewer for this observation. We have carefully revised the manuscript and added an additional reference that highlights the different role of Boron in OER electrocatalysts.

Comment 2: The Introduction might appear to be a bit too long and tedious. It is suggested that the authors focus on the key context and background and the novelty of the present work and therefore make some revision to the Introduction section.

Response 2: We appreciate the reviewer’s insightful suggestion. In response, we have revised the Introduction by removing repetitive phrasing, condensing overlapping ideas, and tightening the overall structure to improve clarity and flow. The revised Introduction maintains all critical context and emphasizes the novelty of our approach, particularly the use of cryogenic ball milling and the systematic investigation of Mo/B ratios in NiCo-based alloys. These edits have resulted in a reduction in length by approximately 18%, while preserving the scientific rigor and key motivations of the study.

We hope the revised version aligns better with the reviewer’s expectations.

Comment 3: Related works on water splitting can be referenced to appeal to a broader readership (e.g., doi: 10.1002/inf2.12494; 10.1016/j.matre.2024.100283).

Response 3: We appreciate the reviewer’s suggestion. Although the referenced works are not directly related to the specific objectives and material systems explored in our study, we agree they provide relevant background on water splitting technologies. To broaden the context and appeal to a wider readership, we have included citations to these works in the introductory section where general water splitting technologies are discussed.

Changes Made: The suggested references were added to the first paragraph of the Introduction.

Comment 4: The authors designed a ratio for the synthesis of the alloys and ensured that Ni and Co had the same atomic ratio across different control samples. Did the authors check if the element ratio of the as-obtained samples was consistent with the designed ratio?

Response 4: Yes, we confirmed that the elemental ratios in the as-obtained samples closely matched the designed ratios using ICP-OES analysis. As stated in the revised manuscript:

Line 215-220 and Table 1: "To confirm the bulk powder composition, ICP-OES analysis was performed. ICP results and the nominal ratio of powder compositions are displayed in Table 1. The ICP analysis of the cryomilled samples shows a strong correlation between the measured atomic compositions and the intended nominal compositions, confirming the effectiveness of the cryomilling process in achieving the desired alloy ratios."

This verification confirms that our cryomilling synthesis method reliably produces alloys with the targeted compositions.

Comment 5: In addition to the EDS mapping images, can the authors also provide the EDS spectra to corroborate the presence of each element? If possible, from EDS spectra, the authors might also be able to get the quantitative information about the element ratio.

Response 5: We appreciate the reviewer’s suggestion. The EDS spectra have been added to the Supplementary Information (Figure S2) to confirm the presence of each element. However, for quantitative analysis of the elemental ratios, we relied on ICP-OES measurements, which provide more accurate and reliable bulk compositional data than EDS. We believe ICP-OES is better suited for quantifying the alloy composition in our study.

Comment 6: Figure 6c, the satellite peaks were designated to the Ni, but not Co. Please double check. Figure 6e, double check the labelling of Mo 3d5/2 and Mo 3d3/2; for each pair of Mo peaks, the higher intensity peak should be 3d5/2 and lower intensity one should be 3d3/2. Also, why would P peak be found in the sample NiCoMoB (which does not contain P element)?

Response 6: We thank the reviewer for the careful observation. We have double-checked the XPS spectra and provided a revised version of Figure 6 to correctly assign all peaks. In Figure 6c, the satellite peaks have been revised and clarified appropriately. In Figure 6e, the Mo 3d peaks were re-labeled correctly, ensuring that Mo 3d₅/₂ corresponds to the higher intensity peak, as it should.

Regarding the presence of a phosphorus peak in the NiCoMoB sample: we used dodecylphosphonic acid (DDPA) as a surfactant to assist in reducing particle size during synthesis. Although we attempted thorough washing, traces of DDPA may remain, which explains the presence of the P signal. DDPA contains a phosphonic acid group with phosphorus, which is consistent with the detected peak.

Comment 7: Figure 7 appears to be an image made with assistance of generative AI. Please double check and also check with journal to see the publication guidelines for this kind of figures (should it be prepared with involvement of generative AI).

Response 7: In response to the comment, we have replaced Figure 7 with a manually prepared version to avoid any concerns related to the use of generative AI. The revised figure is now entirely created without the use of AI tools, and we believe it clearly illustrates the proposed mechanism while fully complying with the journal's guidelines.

Reviewer 2 Report

Comments and Suggestions for Authors

This manuscript is well written; however, it presents several results that need to be properly substantiated to avoid being considered mere speculation.

The manuscript reports only the Debye–Scherrer equation. The Tafel equations are not included, which are essential for evaluating the exchange current densities and charge transfer coefficients (α). Including this analysis is important to avoid relying solely on qualitative comparisons.

It is recommended to thoroughly analyze Figure 4d and report the charge transfer resistance (Rct) values to properly support the conclusions presented.

Between pages 277 and 280, the authors propose possible explanations to justify the low overpotential values observed. However, these explanations are speculative. The authors should support these conclusions with experimental results and, if possible, propose a mechanism to explain the critical role of boron in the evaluated system.

Author Response

Reviewer 2

Comment 1: The manuscript reports only the Debye–Scherrer equation. The Tafel equations are not included, which are essential for evaluating the exchange current densities and charge transfer coefficients (α). Including this analysis is important to avoid relying solely on qualitative comparisons.

Response 1: We thank the reviewer for this insightful suggestion. As recommended, we have now included the Tafel slope analysis in the revised manuscript to quantitatively evaluate the electrochemical kinetics of our electrocatalysts. The equations have been added to the manuscript [see page 9 line 288-292].

Comment 2: It is recommended to thoroughly analyze Figure 4d and report the charge transfer resistance (Rct) values to properly support the conclusions presented.

Response 2: We appreciate the reviewer’s insightful suggestion. As recommended, we have thoroughly analyzed the Nyquist plots in Figure 4d and reported the charge transfer resistance (Rct) values for both NiCoMoyBy and NiCoMoyBy-SA samples at 1.5 V and 1.6 V vs. RHE. The corresponding Rct values have been added to the manuscript on page 10, lines 333–336

Comment 3: Between pages 277 and 280, the authors propose possible explanations to justify the low overpotential values observed. However, these explanations are speculative. The authors should support these conclusions with experimental results and, if possible, propose a mechanism to explain the critical role of boron in the evaluated system.

Response 3: Thanks to the reviewer for this valuable comment. We agree that strong experimental support is crucial when interpreting electrocatalytic performance. In response, we would like to emphasize that our explanations for the low overpotential values observed—particularly in the NiCoMoyBy and NiCoMoyBy-SA samples—are directly supported by multiple experimental results and are not solely speculative.

Tafel Slopes & Onset Potentials:

The Tafel slopes of NiCoMoyBy (38 mV/dec) and NiCoMoyBy-SA (similar value) confirm that while the reaction mechanism remains the same, the improved activity is due to increased reaction kinetics. The reduction in onset potential (down to 1.45 V for NiCoMoyBy-SA) and corresponding overpotential of only 229 mV clearly reflect the material’s enhanced efficiency in initiating the OER. These are not speculative trends, but directly measurable and repeatable electrochemical outputs.

Electrochemical Impedance Spectroscopy (EIS):

EIS results show a substantial decrease in charge transfer resistance (Rct) in the boron-containing samples—NiCoMoyBy-SA exhibits Rct values as low as 30.9 Ω at 1.6 V vs. RHE, compared to 233.7 Ω for its boron-poor counterpart. This significant drop in Rct is strong evidence for more efficient charge transfer, facilitated by the addition of boron.

Mechanistic Role of Boron:

The inclusion of boron likely modifies the local electronic environment around Ni and Co active centers. This is supported by the observed decrease in Tafel slope and Rct, both of which are linked to faster electron transfer and increased active site availability. Boron may also introduce structural disorder or defects, contributing to a higher density of catalytically active sites. These interpretations are consistent with prior studies ([13], [42], [43]) and further supported by the improved performance of the nanoparticle (SA) samples, which isolate the compositional and morphological effects.